



# Carbon Exchange in an Amazon Forest: from Hours to Years

**Matthew N. Hayek[1], Marcos Longo[2], Jin Wu[3], Marielle N. Smith[4], Natalia Restrepo-Coupe[5], Raphael Tapajós[6], Rodrigo da Silva[6], David R. Fitzjarrald[7], Plinio B. Camargo[8], Lucy R. Hutyra[9], Luciana F. Alves[10], Bruce Daube[11], J William Munger[11], Kenia T. Wiedemann[11], Scott R. Saleska[12], and Steven C. Wofsy[11]**

[1] Harvard Law School, Cambridge, MA, United States
[2] NASA Jet Propulsion Laboratory, California Institute of Technology, Pasadena, CA, United States
[3] Biological, Environmental & Climate Sciences Department, Brookhaven National Lab, Upton, New York, NY, United States
[4] Department of Forestry, Michigan State University, East Lansing, MI, United States
[5] Plant Functional Biology and Climate Change Cluster, University of Technology Sydney, Sydney, NSW, Australia.
[6] Universidade Federal do Oeste do Pará, Santarém, PA, Brazil.
[7] University at Albany SUNY, Albany, NY, United States.
[8] Centro de Energia Nuclear na Agricultura, Universidade de São Paulo, Piracicaba, SP, Brazil.
[9] Department of Earth and Environment, Boston University, Boston, MA.
[10] Center for Tropical Research, Institute of the Environment and Sustainability, UCLA, Los Angeles, CA, United States.
[11] Faculty of Arts and Sciences, Harvard University, Cambridge, MA, United States.
[12] Department of Ecology and Evolutionary Biology, University of Arizona, Tucson, AZ, United States.

*Corresponding author*: Matthew Hayek (mhayek@law.harvard.edu)

## Abstract

In Amazon forests, the relative contributions of climate, phenology, and disturbance to net ecosystem exchange of carbon (NEE) are not well understood. To partition influences across various timescales, we use a statistical model to represent eddy covariance-derived NEE in an evergreen Eastern Amazon forest as a constant response to changing meteorology and phenology throughout a decade. Our best fit model represented hourly NEE variations as changes due to sunlight, while seasonal variations arose from phenology influencing photosynthesis and from rainfall influencing ecosystem respiration, where phenology was asynchronous with dry season onset. We compared annual model residuals with biometric forest surveys to estimate impacts of drought-disturbance. We found that our simple model represented hourly and monthly variations in NEE well ($R^2$ = 0.81, 0.59 respectively). Our model also simulated annual NEE well, with exception to 2002, the first year of our data record, which contained 1.2 MgC ha$^{-1}$ of residual net emissions, because photosynthesis was anomalously low. Because a severe drought occurred in 1998, we hypothesized that this drought caused a persistent, multi-year depression of photosynthesis. We did not find evidence to support the common assumption that droughts or disturbances affected this region during 2005 or 2010, nor that the forest phenology was seasonally light- or water-triggered. Our results suggest drought can have lasting impacts on photosynthesis, possibly via partial damage to still-living trees.

## 1. Introduction

The Amazon's tropical forests are pivotal to global climate, containing 10-20% of Earth's biomass (Houghton et al., 2001). Increased emissions of the forest's carbon can accelerate climate change (Betts et al., 2004)



and attention is now focused on how vulnerable this large reservoir of carbon will be to a potentially drier future
climate (de Almeida Castanho et al., 2016; Farrior et al., 2015; Duffy et al., 2015; McDowell et al., 2018).
Characterizing the response of present-day Amazon rain forest carbon balance to climate and drought disturbance is
a necessary step to improving predictions of future vulnerability.

Eddy covariance $CO_2$ flux measurements are a powerful tool for quantifying net ecosystem exchange of

carbon (NEE) (Baldocchi, 2003). NEE is the difference between uptake from gross ecosystem productivity (GEP)
and emission from ecosystem respiration (RE). The magnitudes of these gross fluxes are influenced both by
exogenous environmental conditions such as light, moisture, and temperature (Collatz et al., 1991; Bolker et al.,
1998; Fatichi et al., 2014), as well as endogenous biophysical properties such as canopy structure, phenology, and
community composition (Barford et al., 2001; Melillo et al., 2002; Dunn et al., 2007; Doughty and Goulden, 2008;
Stark et al., 2012; Frey et al., 2013; Morton et al., 2016; Wu et al., 2016).

Partitioning the exogenous and endogenous influences upon eddy covariance NEE is possible using

statistical modeling (Barford et al., 2001, Yadav et al., 2010). To partition influences upon NEE in a 20-year eddy
flux record in a temperate New England forest, Urbanski et al. (2007) used a statistical modeling approach: by
representing hourly NEE merely as response to exogenous meteorology and annually integrating their results, they
concluded that meteorology did not explain the accelerated uptake seen annually integrated NEE. They hypothesized
that residual uptake was due to long-term forest regrowth and succession, a hypothesis that was corroborated by
biometric measurements of increasing canopy foliage and accelerating mid-successional tree biomass accrual. This
novel partitioning framework for NEE has not previously been applied to any tropical forest, in part because long-
term eddy covariance coverage of tropical forests is lacking (Zscheischler et al., 2017). A simple statistical
framework may allow tropical forest $CO_2$ flux measurements to better inform model development and improvement.

On seasonal timescales, tropical evergreen forests undergo endogenous changes in GEP via the phenology

of leaf flush and abscission (Doughty and Goulden, 2008, Restrepo-Coupe et al., 2013). Such seasonal dependency
of productivity has motivated the development of rooting depth and phenology sub-models in DVGMs (Verbeeck et
al., 2011; De Weirdt et al., 2012; Kim et al., 2012). These sub-models have led to complexity in the modeled
mechanisms controlling the GEP seasonal cycle without necessarily improving its fit to measurements. It is
necessary to determine whether these sub-models represent the correct magnitude and timing of the GEP seasonal
cycle after accounting for the integrated hourly response to sunlight.

On interannual to decadal timescales, endogenous changes in forest NEE can arise from disturbance and

recovery (Nelson et al., 1994; Moorcroft et al., 2001; Chambers et al., 2013; Espírito-Santo et al., 2014; Anderegg et
al., 2015). The km67 eddy flux site in the Tapajós National Forest presents a unique opportunity to study the
potential legacy of disturbance caused by drought. This Eastern Brazilian Amazon forest lies on the dry end of the
rainfall spectrum for tropical evergreen forests (Saleska et al., 2003; Hutyra et al., 2005). A severe El Niño drought
in 1997-1998 was followed by disturbance, evidenced by a large and heavily respiring CWD pool in 2001.
Subsequent NEE measurements showed a 4-year transition from a net carbon source in 2002 to nearly carbon-
neutral in 2004 and 2005 (Hutyra et al., 2007). The observed disequilibrium state led researchers to the hypothesis
that RE was high but dissipating, and that the forest will continue to transition into equilibrium, becoming a sink



throughout the decade (Pyle et al., 2008). Conversely, this hypothesis implies that any new disturbance should drive
the forest back into disequilibrium, becoming a source again. We test these predictions using meteorological
records, forest inventories of aboveground biomass (AGB) and CWD, and an additional 3.5 years of eddy flux data,
resumed after a 2.5-year interruption, collected since prior studies.

In this study, we test hypotheses related to controls of NEE on multiple timescales at an Eastern Amazon

rain forest. Specifically, we sought to answer the following questions: (1) how accurately can NEE be modeled
using the mean response to meteorological forcing throughout the entire updated 7.5-year eddy flux record? (2)
What is the seasonal effect upon GEP of canopy phenology? Is phenology itself synchronized with wet/dry
seasonality? (3) Major basin-wide droughts occurred in 1998 before eddy flux measurements began, and were
reported again in 2005 and 2010 (Zeng et al., 2008; Philips et al., 2009; Lewis et al., 2011; Doughty et al., 2015)
during the span of measurements. Can we infer from meteorology, biometric data, and the NEE-model residuals
which basin-wide droughts impacted this particular region? Which NEE component, GEP or R, was perturbed most?
Overall, we statistically partitioned the multiple influences on NEE across timescales from hours to an entire decade
of eddy flux and forest inventory measurements.
**2 Methods**
**2.1 Site Description**

The Tapajós National Forest is located to the southeast of the convergence of the Tapajós and Amazon

Rivers in Pará, Brazil. The forest site is on the dry end of the spectrum of evergreen tropical forests, receiving 1918
mm of annual rainfall and experiencing a 5 month long dry season (Hutyra et al., 2007). The forest has a closed
canopy with a height of roughly 40 m (Stark et al., 2012), emergent trees up to 55 m (Rice et al., 2004), fast turnover
rates with much of the population consisting of small-diameter trees (Pyle et al., 2008). The flux tower that provided
flux and meteorological data is located at km 67 of the Santarém-Cuiabá highway. The tower and site are designated
by site ID "BR-Sa1" in the FLUXNET data system, but are herein referred to simply as "km67".
**2.2 Eddy Covariance Measurements**

Hourly fluxes of NEE were calculated using the sum of hourly turbulent eddy fluxes plus the hourly change

in height-weighted average $CO_2$ concentration in the canopy air column. Our measurements covered two contiguous
periods: one from January 2002 to January 2006 (period 1) and another from July 2008 to December 2011 (period
2). The tower fell in January 2006 when a tree snapped a supporting guy-wire. Measurements resumed in July of
2008 when the tower was rebuilt and equipment repaired. Measurements ceased again in 2012 when electrical
failures damaged measurement and calibration systems. Some data collection has resumed since 2015, although
gaps in this data were much larger than those in periods 1 and 2, precluding calculating annual carbon balance after

2011.





**2.3 Flux Data Processing, Quality Control, and Gap Filling**
Nighttime NEE measurements were filtered for low turbulence. We used a turbulence threshold filter of
$u_*^{Th}$ = 0.22 to ensure consistency with previous studies (Saleska et al., 2003; Hutyra et al., 2008). The absolute
magnitude of nighttime respiration and resulting carbon balance was highly sensitive to the selection of $u_*^{Th}$,
(Saleska et al, 2003; Miller et al., 2004). However, the interannual variability and trend remained the same
regardless of the choice of $u_*^{Th}$. Errors in total annual NEE therefore do not reflect potentially large $u_*^{Th}$ error, and
should be interpreted as errors in the differences between years, not errors in the annual magnitude of the carbon
source/sink.
We used well-established gap-filling models to obtain annual NEE totals. Gross ecosystem productivity
(GEP) was gap-filled using a hyperbolic fit curve between GEP and PAR (Waring et al., 1995). For ecosystem
respiration (*R*), we adapted the method by Hutyra et al. (2007), who calculated missing, filtered, and daytime hours
using 50 $u_*$-filtered nighttime hour bins, instead using a running average of 50 $u_*$-filtered nighttime hours, allowing
us to capture the onset of semiannual seasonal transitions in *R*. Consistent with other tropical forest sites,
temperature was not used in our gap-filling, because temperature variability at tropical forests is low, which results
in weak and insignificant correlations with RE (Carswell et al., 2002). We calculated annual errors as 95% bootstrap
confidence intervals by resampling like-hours with replacement (NEE conditions for the same month, time of day,
and similar PAR conditions), instead of resampling all hourly NEE, so that resampling did not capture diurnal and
long-term nonstationary.
**2.4 Meteorological Measurements**
Meteorological variables measured at km67 included photosynthetically active radiation (PAR),
temperature, and specific humidity. Downward drifts in PAR data due to a degrading sensor were corrected by de-
trending a time series of mid-day PAR observations in the top 95th percentile of each month (Longo, 2014). This
threshold included substantial information about the sunniest hours, throughout which intensity should remain
constant between years for any given month. We scaled the radiation time series using the proportion between the
fitted trend and the initial fitted value. Simultaneous total incoming shortwave radiation measurements allowed us to
partially fill missing periods of PAR data using a relationship derived from linear regression in simultaneously
measured hours ($R^2$ = 0.98).
Rainfall measurements were greatly underestimated at this site because of a faulty tipping bucket rain
gauge. We discarded site-based data and calculated a distance-weighted synthetic hourly rainfall time series from a
network of nearby meteorological stations, with locations ranging from 10 km to 110 km away from km67. More
information on the meteorological network is available in Fitzjarrald et al. (2008). Detailed information about the
subsequent calculations of the synthetic precipitation data set and PAR drift correction are available in Longo

(2014).

Additionally, the Brazil National Institute of Meteorology (INMET) has a station at Belterra, located 25 km
away from km67, with daily precipitation totals dating back to 1971, which were used to corroborate the seasonal



and long-term trends at km67. Altogether there were three data sets: the local tower-based meteorology, the
mesoscale network meteorology data interpolated to km67, and the INMET meteorology, which provided us with at
least two redundant estimates for all meteorological variables at km67.
**2.5 Coarse Woody Debris and Mortality**
To assess how disturbance coincided with changes in NEE, we conducted surveys of coarse woody debris
(CWD). These surveys capture the magnitude and dynamics of the respiring pool of dead tree biomass. Transect
subplots were surveyed in 2001 for pieces greater than 10 cm in diameter (Rice et al., 2004). Bootstrapped
confidence intervals were quantified by resampling subplots totals (n=321) with replacement. Additionally, in 2006,
pieces only greater than 30 cm in diameter were surveyed. Lastly, we conducted an additional CWD survey in 2012
using the line-intercept method (Van Wagner, 1968) throughout all transects for a total length of 4 km to minimize
sampling uncertainty. Bootstrap confidence intervals were quantified by resampling line segment totals (n=40) with
replacement. These two different methodologies have previously produced consistent simultaneous results within
measurement uncertainties, which were 20% larger for line-intercept sampling than plot-based sampling (Rice et al.,

2004).

Because CWD surveys were conducted infrequently, we inferred mortality from aboveground biometry
surveys in 1999, 2001, 2005, 2008, 2009, 2010, and 2011. Trees larger than 10 cm diameter at breast height (DBH)
were surveyed and were converted to biomass using non-species specific equations (Chambers et al., 2001a) based
on sampling previously established protocols for this site (Rice et al., 2004; Pyle et al., 2008). Mortality biomass
was inferred by tallying biomass of dead trees that were alive in the prior survey. Sometimes, trees were missed by
the census surveyors before they could be confirmed dead or were found again. In 2012 we assigned missing trees
that were not later found alive an equal probability of dying in all surveyed years they had been missing (Longo,
2014). We used tree mortality to model CWD over time using a simple box model with a first-order rate equation:
$$\frac{dCWD}{dt} = -kCWD + M \qquad (1)$$
where M is the mortality rate input to the CWD pool (MgC ha$^{-1}$yr$^{-1}$) and k is the decay loss rate of 0.124 yr$^{-1}$. The
loss rate is derived from measurements of respiring CWD in Manaus, Amazonas (Chambers et al, 2001b) and snag
density measurements taken at km67 (Rice et al., 2004). The box model initial condition was the 2001 survey of
total CWD. This model allowed us to assess whether disturbances after 2001 were sufficient to cause an increase in
CWD or whether disturbances after 2001 were minimal and the CWD pool respired and depleted gradually.
**2.6 Empirical NEE Model**
Our low-parameter empirical model represents the mean response of NEE to hourly and seasonal changes
in exogenous meteorology and seasonal changes in phenology throughout the decade. We use our model to diagnose
interannual nonstationarity in model residuals, which correspond to endogenous ecosystem changes in
photosynthesis and respiration rates between years, give or take random measurement error and unaccounted for





model terms. We fit the model to the entire 7.5-year interrupted eddy covariance record of raw, $u_*$-filtered hourly
NEE (NEE$_{obs}$):
$$NEE_{Model} = a_0 + a_1 s_R + \frac{a_2 PAR}{a_3 + PAR} \cdot (1 - k_{pheno} s_{pheno})$$

where NEE$_{Model}$ is the modeled hourly NEE. The models were fit in two steps: first, the two model parameters that
represent $R$, $a_0$ and $a_1$, were first fit to nighttime data, then the remaining three GEP parameters were fit to daytime
data. Parameter $a_0$ is the wet season intercept for $R$. Parameter $a_1$ is an adjustment of the ecosystem respiration
during the rainfall-defined dry season (factor variable $s_R$, defined in detail below). Parameters $a_2$ and $a_3$ are the
Michaelis-Menten light response parameters. We also include a simple scaling factor for endogenous changes in
phenology: a time-varying binary factor variable $s_{pheno}$ represents timing in changes to the intrinsic light use
efficiency (LUE≡1-$k_{pheno}$) within an average seasonal cycle. The purpose of this simplistic scaling factor was to
determine when the timing of endogenous seasonal shifts in LUE that were not explained by light and moisture were
most pronounced.

This forest site has coincident deficits in rainfall and ecosystem RE during the dry season (Saleska et al.,

2003; Goulden et al., 2004) due to desiccation of dead wood, leaf litter, and other substrates for heterotrophic
respiration (Hutyra et al., 2008). To depict this reduced dry season $R$, we set dry season $s_R$≡1 and wet season $s_R$≡0,
fitting $a_1$ to the mean dry season $R$. We defined the dry season onset as the period during which rainfall is below
50mm per half-month and the wet season onset as the first in a series of 3 or more semi-monthly periods with
rainfall greater than 50mm, allowing for sporadic dry season downpour and ensuring that there is not more than one
dry season per year. Although $a_1$ does not vary across years, our meteorologically-defined $s_R$ permits the duration of
the dry season to vary interannually. A longer dry season in a given year would therefore result in less RE (more net
uptake) when NEE$_{Exo}$ is integrated over that full year.

We tested three different seasonal timings for the phenology factor variable: (1) $s_{pheno}$ ≡0 year-round (no

phenology), (2) $s_{pheno}$ ≡1 during the dry season and $s_{pheno}$ ≡0 during the wet season, and (3) $s_{pheno}$ ≡1 during the peak
of leaf flush (June 15 to Sept 14) (Hutyra et al., 2007) and $s_{pheno}$ ≡0 all other times of the year. In scenario 2, the
timing of phenology varies interannually, but in scenarios 1 and 3, modeled phenology does not differ between years
and therefore does not influence interannual variability in modeled GEP or NEE.

After subtracting hourly NEE$_{Model}$ from NEE$_{obs}$, the annually integrated residuals reflect changes in the

ecosystem's efficiency irrespective of the aggregate response to meteorology, plus or minus random error and
unaccounted for meteorological controls. Upper-level soil moisture, for instance, may exert some controls, but is not
included in the model because it was insignificantly associated with GEP or RE at this deep-rooted tropical site.
Examples of a change in intrinsic ecosystem efficiency may occur in the aftermath of a drought, during which leaf
stomates close, causing the ecosystem to sequester less $CO_2$ per unit incident PAR than average, or a storm inducing
widespread mortality and a pulse of CWD during which RE would be higher than average for a given season or
year. In both scenarios, we would expect residuals to be positive during or after the event, because the ecosystem
would sequester less and emit more $CO_2$ relative to other years. To assess which aggregated annual residuals were



significantly different from zero, we quantified 95% confidence intervals in annual NEE residuals due to random
error using bootstrapping (Section 2.3).

We partitioned both $NEE_{obs}$ and $NEE_{Model}$ into RE and GEE (GEE = -GEP, to keep the same sign

convention as eddy flux NEE) to determine which of the two components were more adequately represented by our
model. For observations of NEE, R, and GEE, we used hours during which a direct $u_*$ filtered measurement of NEE
occurred. Observations of RE were nighttime hours during which NEE was measured; observations of GEE are
daytime hours during which the 50-hour running average RE was subtracted from measured NEE. Partitioned GEE
is not a direct observation, but represents the lowest-parameter approximation of a direct measurement. Our
GEE/RE results are limited by not accounting for partitioning bias.
**3 Results**
**3.1 Eddy Covariance Measurements of $CO_2$ Fluxes**

Coverage of hourly NEE was substantial for both periods in the total eddy covariance record. After quality

control and outlier detection, period 1 (2002-2006) had 80% and period 2 (mid 2008-2011) had 75% data coverage
for all hours. Filtering for $u_*$ below the threshold of 0.22 m/s left 48% and 42% coverage of period 1 and 2
respectively. NEE has a strong diurnal cycle, with a mean diel range of 25.05 $\mu$mol m$^{-2}$ s$^{-1}$. The range of the mean
seasonal cycle is 2.46 $\mu$mol m$^{-2}$ s$^{-1}$, or 10% of the mean diel range.

Annual totals of NEE are presented in Fig. 1. For period 1, the first four years, annual NEE is similar to that

reported previously by Hutyra et al. (2007). The previously reported trend remains: a moderate source in 2002 of 2.7
MgC ha$^{-1}$ yr$^{-1}$ (±0.5 95% bootstrap confidence intervals) tapering off to nearly carbon neutral totals in the following
years, within confidence limits, of 0.5 (±0.6) MgC ha$^{-1}$yr$^{-1}$ in 2004 and 0.2 (±0.6) MgC ha$^{-1}$yr$^{-1}$ in 2005. Slight
changes in the gap-filling and quality control resulted in insignificant changes to the annual totals between studies.
During the three subsequent years that comprise period 2, 2009-2011, the forest returned to a moderate source of
carbon, with a range of 1.8 ± 0.6 MgC ha$^{-1}$yr$^{-1}$ in 2010 to 2.5 ± 0.5 MgC ha$^{-1}$yr$^{-1}$ in 2009. We examined
measurements of rainfall, coarse woody debris (CWD), and aboveground biomass (AGB) for indications of drought
or other disturbance during 2002-2011 to explain these patterns seen in annual NEE totals.
**3.2 Meteorological Measurements and Drought**

We examined our distance-weighted interpolated estimate of km67 rainfall for trends and droughts. Our

precipitation estimate was consistent with previous estimates of precipitation for this site and region, with a
minimum of 1595 mm in 2005 and maximum of 2137 mm in 2011 (Saleska et al., 2003; Nepstad et al., 2007).
While 2005 annual precipitation was a minimum, no previous groundwater deficits in carbon exchange, latent heat
flux, or sensible heat fluxes were observed during this year (Hutyra et al, 2007). Our measurements did not indicate
that any drought occurred during or immediately preceding period 2 of NEE measurements.  In fact, period 2 annual
rainfall totals increased on average by 20% relative to period 1. The dry season in 2009 was longer than average,



lasting 6 months (Fig. 2a). Mean annual radiation was expectedly anti-correlated with annual rainfall. Accordingly,
period 2 experienced 4% less mean annual PAR than period 1.

Our synthetic decade-long rainfall record corresponded closely with the nearby INMET Belterra

measurements, although INMET Belterra had on average 220 mm of rainfall more per year, likely due to differences
in circulation and convection between the km67 forest and Belterra pasture land surface (Fitzjarrald et al., 2008).
Annual rainfall totals throughout the decade of eddy flux measurements 2002-2011 lay well within the historical
variability of annual rainfall since 1972, which experienced a range of 974 to 3057 mm of annual precipitation (Fig.
2b). The second and third lowest annual precipitation totals occurred during 1997-1998, which were 1391 and 1218
mm respectively, during a major El Niño event, which persisted from June of 1997 to June of 1998 (Ross et al.,
1998) and corresponded with a 9 month long dry season, the longest in the historical record.
**3.3 Coarse woody debris and mortality**

We examined measurements of CWD over time to assess whether a disturbance might have impacted the

period 2 carbon balance. Compared to CWD stocks in 2001 of 48.6 (± 5.9) MgC ha$^{-1}$, CWD stocks in 2012 were
significantly lower at 30.5 MgC ha$^{-1}$ (± 7.4) (Fig. 3). Errors in the 2012 pool were 25% larger. The larger magnitude
of error is consistent with higher uncertainty for line-intercept sampling relative to area-based sampling at the TNF
(Rice et al., 2004). Because CWD measurements were sparse in time, we included an additional measurement in
2006 of large CWD, with diameter greater than or equal to 30 cm, totaling 20.8 ± 12.8 MgC ha$^{-1}$. We compared this
measurement with similarly sized CWD from other surveys (Fig. 3). Total large CWD was 25.7 ± 11.4 MgC ha$^{-1}$ in
2001, and 19.8 ± 11.9 MgC ha$^{-1}$ in 2012.  Differences in large CWD between 2001 and 2006 and between 2006 and
2012 are small relative to their uncertainties, but they still show a qualitative downward trend over time.

A box model of CWD (Eq. 2) allowed us to estimate the transient behavior of the CWD pool throughout

years in which it was not directly measured (Fig. 3). The CWD mortality input rates $M$ were derived from forest
inventory surveys. The box model shows no large spikes from mortality events outweighing the respiration rate, and
its derivative is negative throughout time, predicting a continuously depleting CWD pool.  The box model estimate
for 2012 CWD is 26.2 MgC ha$^{-1}$, and lies well within the uncertainty of the concurrent 2012 measurement. We see
no evidence via increased CWD that disturbance has occurred since the start of measurements.

Assuming that the large initial CWD pool arose from a past disturbance, hypothetically following the 1997-

1998 El Niño drought, we ran the CWD box model (Eq. 2) backward in time to estimate the magnitude of such a
disturbance. Because the CWD measurement was made in July of 2001, we calculated the box model CWD value to
the end of the El Niño drought in June 1998 using the same respiration rate, $k$, and the mean mortality, $M$, for all
surveys, and applied this rate to the mean and 95% bootstrapped confidence intervals of the 2001 measurement (48.6
± 5.9 MgC ha$^{-1}$). Our estimate of the CWD pool immediately following the drought was thus 63.7 ± 8.1 MgC ha$^{-1}$.
Subtracting the 2012 measurement of 30.2 ± 7.3 MgC ha$^{-1}$ from this number, which is our best estimate of
equilibrium CWD that may have existed before the 1997-1998 El Niño drought, we estimate drought-induced
mortality to be 33.5 ± 15.4 MgC ha$^{-1}$, or 12-31% of present AGB.



### 3.4 Empirical NEE Model


Optimized parameter values for our model are included in Table 1. Our model predicted 81% of the
variance in observed hourly NEE, and captured 94% of the amplitude of the diurnal cycle. Modeled hourly
variability frequently captured the difference in magnitude in NEE between high and low uptake events (Fig. 4).

### 3.4.1 Seasonal patterns in NEE


The best-fitting LUE parameterization for seasonal phenology was that in which the phenology factor
variable $s_{pheno} \equiv 1$ during the peak of leaf flush (June 15 to Sept 14) and was asychnronous with the dry season (Table
2). Daily averages of the hourly residuals over a mean seasonal cycle highlight the performance of the various
phenology parameterizations (Fig. 5). Removing $s_{pheno}$ results in consistently positive residual NEE from June 15 to
September 14, indicating that the model over-predicts uptake during this time (Fig. 5a). Our final model, however,
simplistically corrects for this positive anomaly and by downscaling the hourly PAR response by a single value (1-
$k_{pheno}*s_{pheno} = 0.84$) during the June-September time period, which only partially overlaps with the dry season (Fig.
5b). Although the phenomena controlling this transition have a gradual, periodical seasonal effect, apparent in the
residuals, our simplistic, low-parameter phenology representation was chosen for parsimony. While the seasonal
timing of respiration, $a_1$, varied by meteorological inputs (semi-monthly total rainfall <50 mm), we could not
identify a similar seasonal meteorological trigger for phenology and therefore used set calendar dates.
Our model predicted monthly mean NEE well ($R^2$=0.59 across all months). Part of the remaining variability
was explained by random measurement error: bootstrap 95% confidence intervals of monthly mean NEE had an
average range of 1.07 $\mu$mol m$^{-2}$s$^{-1}$, representing 47% of the mean NEE seasonal cycle's range. The model slightly
over-predicted the mean seasonal cycle's magnitude, although well within the model and measurement interannual
variability (Fig. 6). The model attributed the greatest sink to October, because (1) October rainfall was low enough
each year to be classified as part of the dry season, (2) PAR was consistently high due to sunny conditions after the
dry season onset, and (3) the phenology scaling factor (1 - $k_{pheno}*s_{pheno}$) returned to 1 after Sept 14, increasing the
October LUE and pushing the carbon balance further towards a sink.

### 3.4.2 Interannual Variability in Modeled NEE Residuals


Including meteorological controls of NEE allowed us to disaggregate hourly and seasonal effects from
long-term changes in forest's ecological efficiency. In 2002, there were a total of 1.2 MgC ha$^{-1}$yr$^{-1}$ of excess
emissions unaccounted for by the modeled mean response to meteorology (Fig. 7a). Importantly, all other years are
not significantly different from zero within random measurement error, represented by 95% bootstrap confidence
intervals, indicating that these years are well predicted by meteorological variability, including the relatively higher
emission/lower uptake in period 2 (Fig. 1). On average, period 2 saw a 20% increase in annual precipitation relative
to period 1. Abbreviated dry season lengths and lack of radiation from increased cloudiness in period 2 resulted in
less modeled net uptake relative to period 1.





We partitioned observed and modeled NEE into RE and GEE. Interannual variations in RE were accurately

represented as changes in wet and dry season length (Fig. S1). The range in annual residual RE is therefore small

compared to that of annual residual GEE (Fig. 7b). In 2002, mean model GEE had 0.85 $\mu$mol m$^{-2}$s$^{-1}$ more uptake

than observations. Therefore, the 1.2 MgC ha$^{-1}$y$^{-1}$ residual emissions in 2002 were more likely due to anomalously

low photosynthesis rather than high $R$.

**4 Discussion**

**4.1 NEE Interannual Variability**

Annual totals of measured NEE exhibited an unpredicted trend: despite previous hypotheses that the years

after period 1 would continue to trend downward towards more uptake (Hutyra et al., 2007; Pyle et al., 2008), the

ecosystem returned to a moderate carbon source in all three years of period 2 (Fig. 1). The surprising finding of the

period 2 source led us to examine whether the interannual variability could be explained by exogenous changes in

climate or an endogenous biophysical change. We developed the model selection framework to partition these two

sources of variability to the best of a statistical model's ability.

Our model represented NEE well across a variety of timescales (Figs. 4, 5, 7). On yearly timescales,

interannual differences in NEE$_{Model}$ were due to exogenous meteorology, as phenology did not vary interannually.

The model predicted annual NEE accurately within 95% confidence limits of random measurement error for 6 out of

7 years (Fig. 7a), including period 2, during which the forest returned to a carbon source (Fig. 1). The model

representation of the period 2 source was due to lower radiation and higher rainfall relative to period 1, consistent

with findings of light-limitation in Amazon forests derived from satellite observations of climate and vegetation

activity (Nemani et al., 2003).

The overall magnitude of the carbon source/sink, however, was highly sensitive to the choice of $u_*$ filter,

consistent with previous findings (Saleska et al., 2003; Miller et al., 2004; Hayek et al., 2018). We therefore applied

a novel correction to the long-term magnitude of NEE that is independent of the $u_*$ filter (Hayek et al., 2018), which

indicated that the ecosystem may in fact be a slight sink, but that the interannual variability, which our model

represents, remained the same (Fig. S2). The overall magnitude of the carbon source/sink therefore does not affect

or results, which concern the variability between years. The least net uptake still occurred in 2002, from which NEE

remained insignificantly different in 2009 and 2011.

The model overestimated GEP in 2002, but predicted RE well (Fig. 7b; Fig. S1). These findings modify a

previously established hypothesis that legacy effects of a prior drought disturbance increased NEE in 2002 via

increased R$_{CWD}$ and related pathways of decomposition (Saleska et al., 2003; Rice et al, 2004; Hutyra et al., 2007;

Pyle et al., 2008). Although we found that R$_{CWD}$ was in fact higher in 2002 than 2005, this difference accounted for

only 0.2 $\mu$mol m$^{-2}$ s$^{-1}$ (Fig. 3) of respiration. Changes in annual R$_{CWD}$ therefore explain the small differences in

annual RE (Fig. S1), but inadequately account for the full 1.3 $\mu$mol m$^{-2}$ s$^{-1}$ (2.4 MgC ha$^{-1}$yr$^{-1}$) difference in NEE

between these years (Fig. 1; Fig. 7). Our model therefore over-predicted photosynthetic uptake in 2002. It remains



just as likely that a prior drought disturbance increased NEE in 2002, but our model results suggest that the legacy
impacts on photosynthesis were greater than impacts on R.
We examined the possibility that a systematic high bias in 2002 PAR could result in an over-prediction of
2002 GEP and erroneously cause a positive 2002 residual. We found that PAR was appropriately drift-corrected by
corroboration with $R_{net}$, which was not affected by drifts. Additionally, we note that rainfall inputs this year were not
atypical in 2002 relative to 2003-2005 (Fig. 2). We examine the possibility that 1998 drought-based disturbance
impacted forest GEP through 2002 in section 4.4.

### 4.2 Implications for the temporal and spatial heterogeneity of droughts

Site-specific precipitation records mirror the large-scale regional interannual variability in Eastern Amazon
rainfall. In the historical precipitation data from Belterra, a major drought was apparent during the 1997-1998 El
Niño, marked by a 9-month long dry season and two consecutive years of annual rainfall below 1500 mm (**Fig. 2b**).
The 40-year historical record had a larger envelope of annual rainfall than that of the last decade alone, implying
that rainfall variability during our ecosystem measurements was within historical variability.
Previous reports of 21[st] century droughts in this region are inconsistent. Lewis et al. **(**2011**)** show that water
deficits during the 2010 drought were minimal in the Eastern Amazon region, consistent with our findings.
However, Doughty et al. **(**2015**)** report ubiquitous detrimental effects of the 2010 drought basin-wide. Doughty et al.
**(**2015**)** report a region of a drought-induced -3 MgC ha[-1] GEP anomaly overlying the Tapajos forest in 2010. Our
results contradict these findings: we did not find anomalously low water inputs, nor a concurrent GEP or NEE
anomaly (Fig. 7b), in 2010. Additionally, Zeng et al. **(**2008**)** claim that North Tropical Atlantic warming in the dry
2005 Jul-Oct quarter led to rainfall reductions everywhere in the Amazon, a result not borne out by our precipitation
analysis. The two supposedly basin-wide droughts in 2005 and 2010 did not appear to affect the region in which this
particular site lies. Measurements and empirical modeling of CWD over time support this finding because no interim
disturbances were detected between 2001 and 2011 (Fig. 3). The spatial extent and severity with which a more
recent 2015-2016 El Niño drought impacted Amazon forests, however, remains to be precisely quantified.

### 4.3. Seasonal Timing of Phenology

The model parameterization contained a seasonal decrease in respiration ($a_1$) that was synchronous with the
dry season, and phenological LUE decrease to GEP ($1-k_{pheno}$) that was asynchronous with the dry season (Eq. 5;
Table 2). Evidence from previous studies at the TNF suggests that changes in phenological LUE result from carbon
allocation shifting from stem allocation to the turnover and production of new leaves (Goulden et al., 2004)
supporting the prevailing hypothesis that tropical trees have been selected to coordinate new leaf production ahead
of dry seasonal peaks of irradiance (Wright and van Schaik, 1994). Seasonal changes in LUE are well explained by
canopy leaf age and demography both at this site and at a comparatively wetter forest site in Manaus, showing good
agreement with a model informed by camera and trap-based observations of leaf flushing and shedding (Wu et al.,
2016). Our single mid-year parameter simplistically up-shifts the trough in a more continuous seasonal oscillation
between low and high LUE (Fig. 5). Without independent variables explaining the seasonal oscillation, a model that




381 corrected for this continuous pattern would be of a higher parameter count and therefore result in over fitting

382 without any additional explanatory power of the effects of phenology on interannual variability.

383   The seasonally asynchronous nature of phenology-mediated LUE establishes a middle ground in debates

384 over whether the Eastern Amazon canopy is enhanced or "greens up" during the dry season (Huete et al., 2006;

385 Myneni et al, 2007; Samanta et al., 2012; Morton et al., 2014; Bi et al., 2015; Guan et al., 2015; Saleska et al.,

386 2016). Changes to the canopy's LUE do indeed occur, but not synchronously with the dry season at our site (Fig. 5).

387 The GEP seasonal cycles at additional evergreen Amazon forest sites are not well described by sunlight alone

388 (Restrepo-Coupe et al., 2013). Averaging over seasonal windows is therefore likely to miss a potential inter-seasonal

389 depletion and enhancement of canopy LUE if additional regions of evergreen Amazon forest similarly exhibit

390 seasonally asynchronous phenology.

391   Interannual variation in phenology is represented mechanistically in phenology and LUE sub-models,

392 which have been optimized using km67 eddy flux data, but nonetheless fail to reproduce the observed mid-year GEP

393 decrease at this site. Kim et al. (2012) present a light-triggered phenology scheme, which assumes higher modeled

394 leaf turnover rates and higher maximum leaf photosynthesis during the dry season, and hence produced higher dry

395 season GEP. Their model produced leaf flushing rates that lagged behind observations, and contradicted

396 observations that light-controlled GEP decreases mid-year at km67 (Fig. 5). Another phenology scheme has been

397 developed by De Weirdt et al. (2012), which attributes excess leaf allocation to the turnover of new, more efficient

398 leaves, but nevertheless over-predicted mid-year GEP at km67 relative to their prior model. Wu et al. (2016a), on

399 the other hand, successfully represent the GEP seasonal cycle using their model of leaf age and demography, but

400 relied on observations of canopy leaf fluxes. Their model, however, does not provide a mechanism for the controls

401 on their seasonal timing. Therefore, until an accurate trigger for seasonal leaf shedding and flushing can be

402 identified, models that mechanistically represent phenology are primed to make erroneous predictions about the

403 interannual and long-term consequences of changing seasonal lengths for the Amazon carbon balance.

404 **4.4 Implications for Impacts of Drought**

405   CWD measurements from the km67 site suggest that there was major disturbance before measurements of $CO_2$

406 eddy fluxes began, but that no impactful disturbance occurred at this site between 2002 and 2012. Three years after

407 the 1998 drought, there was a large pool of CWD (48.6 MgC ha$^{-1}$ in 2001), which was significantly depleted by

408 2012, and which respired faster than it could accrue additional necromass from mortality (Fig. 3). Our

409 meteorological and biometric results, in tandem with significant annual model residuals in 2002 (Fig. 7) are

410 consistent with the hypothesis that a drought-disturbance persistently affected forest GEP.

411   Identifying the cause of the reduced 2002 GEP is beyond the scope of this statistical modeling study. It is

412 possible that the 1997-1998 El Niño drought not only killed entire trees, but also damaged living trees through

413 hydraulic failure and partial limb death, affecting canopy photosynthesis for subsequent years. An analysis of over

414 1000 temperate forest census sites suggests that recovery of live tree biomass accumulation may be delayed by up to

415 four years after drought (Anderegg et al., 2015). Following the 2005 and 2010 western droughts, findings from

416 forest inventories (Brienen et al., 2015) and remote sensing (Saatchi et al., 2013), suggested that legacy effects from



tropical forest droughts can also persist for four years or more. Drought cavitation due to xylem embolisms reduces
hydraulic conductivity leading to whole tree mortality (Choat et al., 2012), initiating a classic disturbance-recovery
scenario in which felled trees generate canopy gaps for early successional seedlings and saplings to immediately
capitalize on newly available light, causing $CO_2$ sources to approximately balance sinks (Chambers et al., 2004).
However, cavitation is also known to cause branch dieback in still living trees (Koch et al., 2004), reducing canopy
foliage partially but not completely forfeiting light resources to the understory. Drought-induced limb diebacks
therefore potentially prolong forest recovery relative to immediate disturbances such as windfall. We hypothesize
that partial drought damage to surviving trees can persistently affect whole-forest photosynthesis. Our findings, that
a 1997-1998 drought-disturbance was followed by reduced photosynthesis in 2002, emphasize the need to better
mechanistically understand multi-year legacy impacts following droughts in evergreen Amazon forests.
**5 Conclusions**
The decade-long record of eddy flux at km67 in the Tapajós National Forest demonstrated surprising trends
in 7.5 years of measured NEE. Our simple, low-parameter empirical model could represent interannual differences
in NEE as integrated continuous responses to changes in meteorology, with exception to the first year, suggesting
that increased moisture and decreased sunlight, not an interim disturbance, were responsible for the elevated period
2 carbon source. Although overall magnitude of the carbon source/sink was highly sensitive to the specific choice of
$u_*$ filter, the interannual variability, which was predicted by the model, remained the same. Contrary to some reports,
no major drought was apparent in concurrent rainfall records, nor was a major concurrent disturbance apparent in
biometry surveys of this site from 2001 through 2011.
Our model represented a seasonal mid-year decline in GEP. Our representation of phenology follows set
calendar dates, and cannot distinguish between various hypotheses concerning the environmental trigger for
seasonal leaf shedding and flushing. DVGMs and other numerical simulation ecosystem models that represent
phenology as a response to light-triggered leaf flushing or root water constraints do not tend to represent the
seasonal cycle of GEP accurately and are therefore in danger of over-predicting the future response of
photosynthesis to longer dry seasons resulting from climate change.
Our finding that reduced photosynthesis, not increased respiration, contributed to the high NEE source in
2002 modifies the previous hypothesis that the 1997-1998 El Niño drought disturbance affected NEE via respiration.
Our findings that photosynthesis was disproportionately affected supports a corollary hypothesis, consistent with
regional and global-scale forest biometric studies, that partial drought-induced damage to still-living trees can
impact whole-ecosystem photosynthesis adversely for multiple years (Anderegg et al., 2015; Brienen et al., 2015).
In order to understand how drought-disturbance uniquely impacts forest recovery, observational studies and plot-
based manipulation experiments are needed in conjunction with models. Such future research is needed to determine
the return times for droughts at which persistent forest biomass loss and collapse may occur.



**Acknowledgments and Data**

This work was supported by funding from a National Science Foundation PIRE fellowship (OISE
0730305) and a U.S. Department of Energy grant (DE-SC0008311). The eddy flux data used in this study are
available online via the Lawrence Berkeley Laboratories (LBL) Ameriflux network database at
http://ameriflux.lbl.gov/sites/siteinfo/BR-Sa1.

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




**Tables and Figures**

| Model Parameters | | | | | hourly $R^2$ | monthly $R^2$ |
|---|---|---|---|---|---|---|
| $a_0$ | $a_1$ | $a_2$ | $a_3$ | $k_{pheno}$ | | |
| 9.43 | 1.32 | 39.2 | 760.9 | 0.164 | 0.81 | 0.59 |

**Table 1. Model parameter values and $R^2$ fit. Parameters have the following units: $a_0$, $a_1$, and $a_2$: μmol-$CO_2$ m$^{-2}$ s$^{-1}$; $a_3$:**
**μmol-photons m$^{-2}$ s$^{-1}$; k$_{pheno}$: unitless.**

| $s_{pheno}$ timing | $k_{pheno}$ | hourly $R^2$ | monthly $R^2$ |
|---|---|---|---|
| None | - | 0.80 | 0.33 |
| Dry Season | 0.117 | 0.80 | 0.32 |
| June 15 to Sept 14* | 0.164 | 0.81 | 0.59 |

**Table 2. $k_{pheno}$ parameter values and hourly and monthly model fit associated with various seasonal timings of the**
**phenology factor variable $s_{pheno}$. *Final model parameterization.**

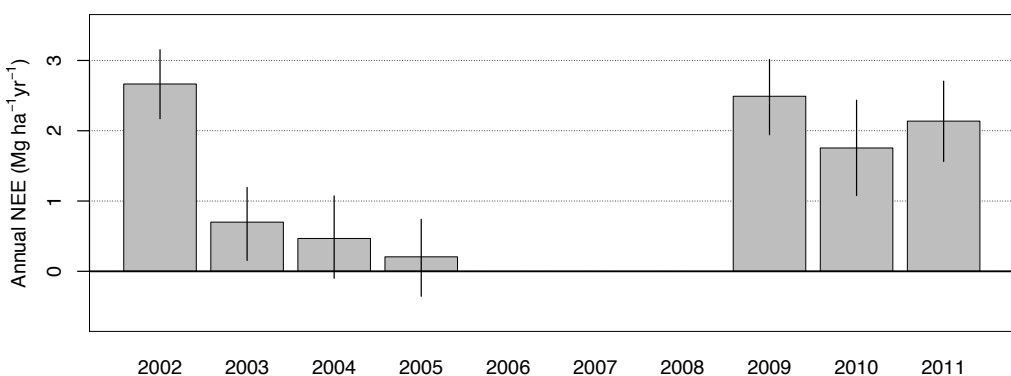

**Figure 1. Annual sums of NEE in kg/ha/year. Error bars are 95% confidence intervals. Positive values indicate a source of**
**$CO_2$ to the atmosphere.**




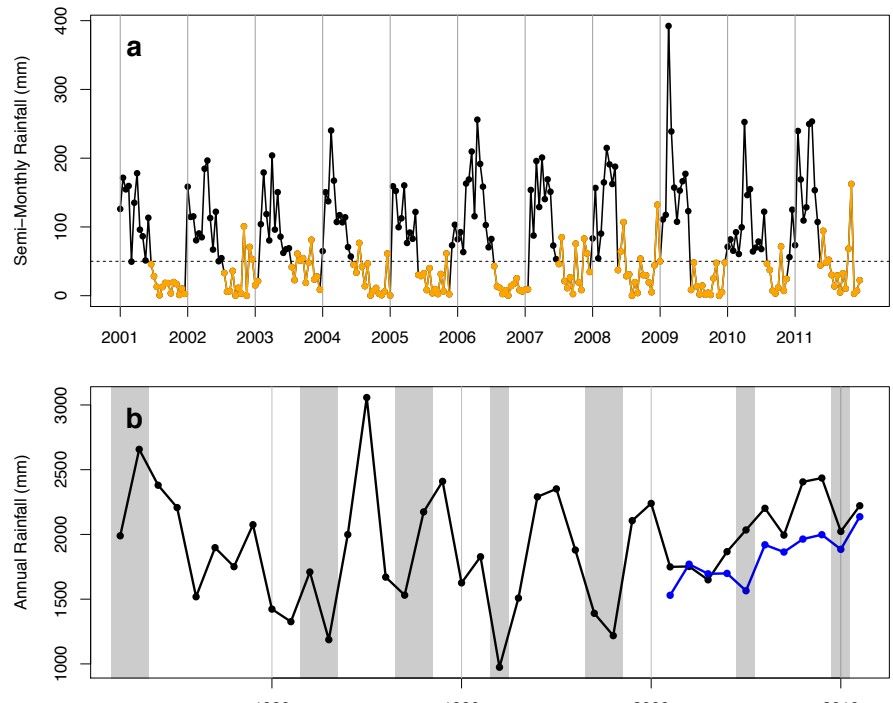

**Figure 2. (a) Semi-monthly dry season rainfall totals for wet season (black) and dry season (orange). Hourly rainfall was estimated by objective analysis (Eq. 1) from meteorology stations nearby km67. The horizontal dashed line shows the dry season threshold of 50 mm per half-month. (b) Yearly totals of rainfall from Belterra INMET station (black), 25 km away from km67, and km67 rainfall estimated by objective analysis (blue). Recent El Niño anomalies (grey shaded areas) coincide with droughts in the 1990s but not in the 2000s (blue points) at this site, when annual rainfall was within the long-term historical variability.**




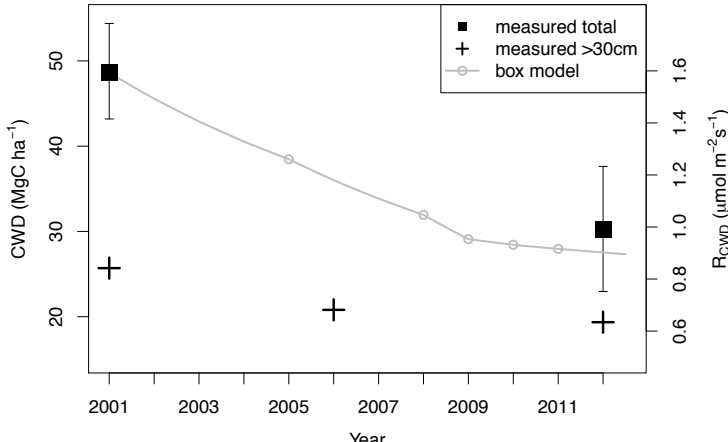

754

**Figure 3. Measurements of total CWD (black squares with 95% bootstrapped CI error bars) and subsets of CWD ≥30 cm**
**diameter (black crosses) show a decrease over time. CWD box model (grey line) also shows a gradual decrease in CWD**
**over time. The initial condition is the 2001 measurement of CWD; source is input from mortality inferred by biometry**
**census (census times represented by grey circles); sink is an empirical respiration rate of 0.124 yr⁻¹ [Pyle et al., 2008]. Left**
**axis shows the CWD respiration flux (R_CWD) corresponding to the equivalent amount of CWD on the right axis.**

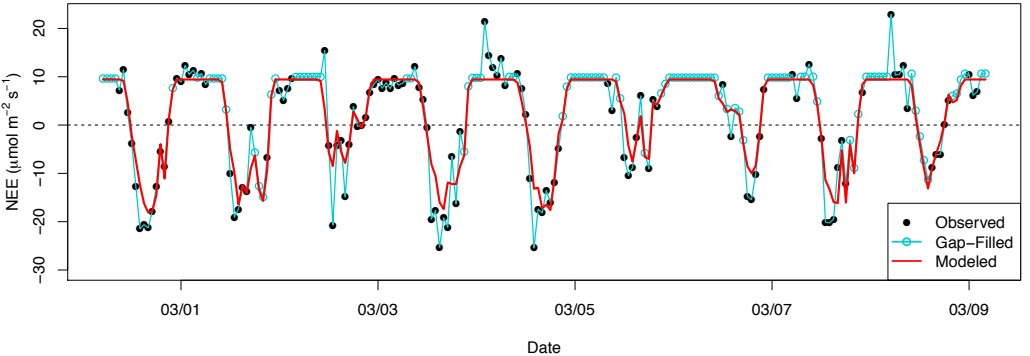


**Figure 4. Sample time series of NEE_obs and NEE_Model for 9 days of the wet season in 2008. Pearson correlation coefficient**
**between NEE_obs and NEE_Model is R=0.90 over the entire 7.5 year time series.**





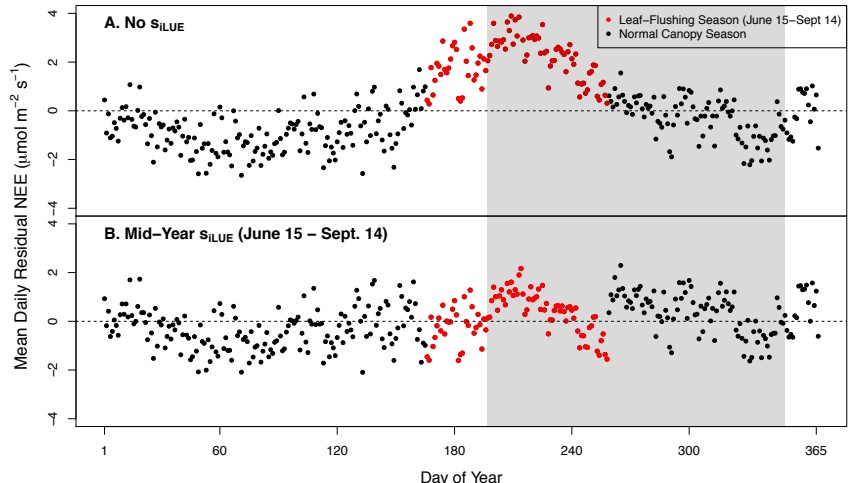


**Figure 5. Mean daily data-model residuals averaged over all 7.5 years: (a) lacks an adjustment for phenological change in**
**LUE. Leaf-flush period only partially overlaps the dry season (grey shaded area). (b) The best-fitting parameterization of**
**the model contained a mid-year phenology scaling factor (1-$k_{pheno}$*$s_{pheno}$ = 0.84; Table 2), which was asynchronous with**
**the dry season (red points).**

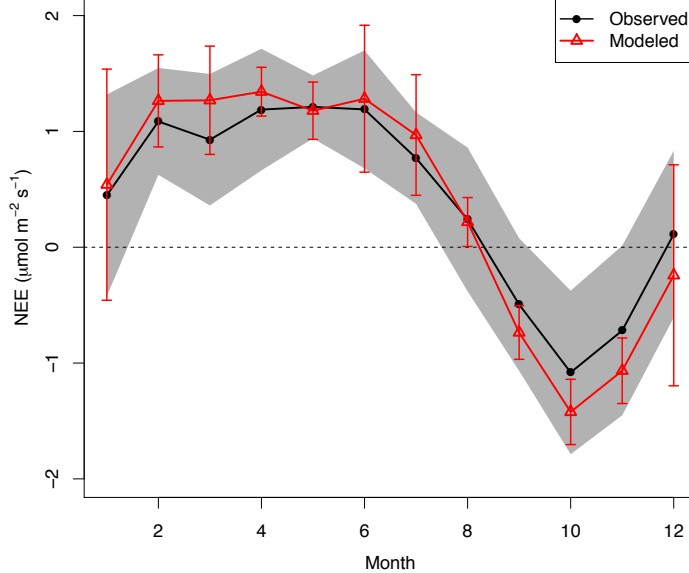


**Figure 6. (a) Mean seasonal cycle of NEE$_{obs}$ (black dots) and NEE$_{Model}$ (red triangles). Grey shaded areas are standard**
**deviations of interannual variability for the mean NEE$_{obs}$ for each respective month. Error bars are standard deviations**
**of the interannual variability in monthly mean NEE$_{Model}$.**






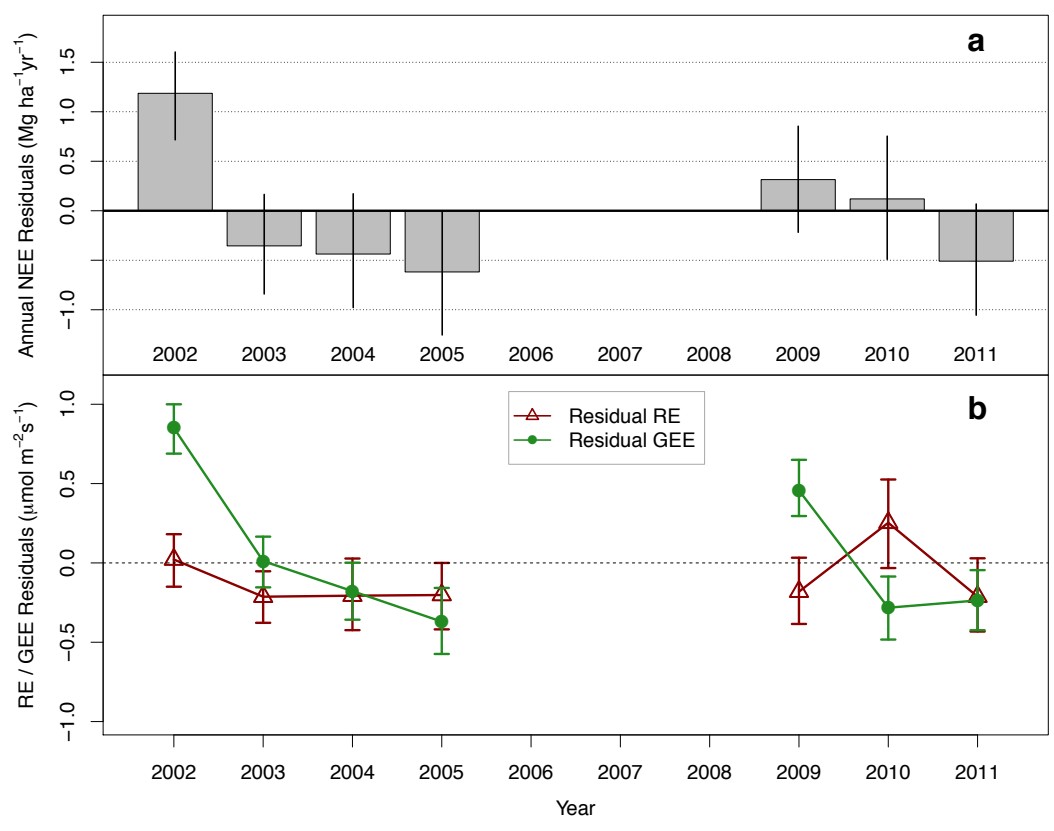


**Figure 7. (a) Annually summed model residuals. Error bars are 95% bootstrapped confidence intervals. Annual residual**
**NEE in 2002 is statistically different from 0 within random NEE measurement error; all other years are not. (b) Residuals**
**of model representation of partitioned GEE (dark green circles) and RE (dark red triangles).**