# Peer review of "Carbon Exchange in an Amazon Forest: from Hours to Years 1"

_Biogeosciences, 2018_

## Referee Comment (RC1) · Anonymous Referee #1 · 17 Jun 2018

This paper investigates the influences of climate, phenology, and disturbance to NEE across various timescales in an evergreen Eastern Amazon forest using a statistical model to represent eddy covariance-derived NEE. What I concerned is the research questions are not clear and lack of storyline through the manuscript. I am curious to know what environmental drivers are responsible for the hourly, seasonal and interannual variability of NEE, respectively, and how many are the relative contributions to the variability of NEE between exogenous changes and endogenous biophysical changes.

Throughout the abstract, it should be more quantitative in nature. For example, how many are the relative influences of climate, phenology, and disturbance to NEE across various timescales?

Line 46-49 please cite the references for each driver respectively.

[Figure]

Line 81-82 did Urbanski et al. (2007) answer this question?

Please double check and use the same term: R/RE.

Line 80-89 what are relationships among these three questions? Each question looks individual.

Line 94 There was 1918 mm of annual rainfall. Why did it have a 5 month long dry season?

Line 91-98 Please describe more about the climate condition (temperature), species composition, vegetation/soil type, forest age, water table depth etc.

Line143-145 did you create the correlation and verified each other for these three data sets?

Line 165-170 did you make any validation for predicted CWD from the box model?

Line 172 what is the low-parameter empirical model?

Line 222-225, these sentences should be placed in Methods, rather than Results.

Line 227-230 since Hutyra et al. (2007) has reported these results, what is the sense of your results here? Same problem with line 238-239.

Line 271 Were there any disturbance for 1998-2000?

Line 762 how about the dry seasonal? Why only 9 days, rather than 90 days?

The results section is pretty confusing. Please organize the idea logically. As your title: 'Carbon Exchange in an Amazon Forest: from Hours to Years', what environmental drivers are responsible for the hourly, seasonal and interannual variability of NEE, respectively? How many are the relative contributions to the variability of NEE between exogenous changes and endogenous biophysical changes?

How did you define the hourly, seasonal and interannual variability of NEE?

Line 325 what is the R2 between modeled and observed annual NEE?

It is also difficult to follow the Discussion section. What are the relationships between these four subtitles? They look pretty individual. Did you discuss the hourly and seasonal variability of NEE? Why don't you focus on explaining your results instead of talking about implications so much? The reader is really curious about your findings.

---

## Referee Comment (RC2) · Anonymous Referee #2 · 11 Jul 2018

Hayek and others explore the net ecosystem exchange of carbon and its components over multiple years in a forest in the eastern Amazon. The simple model that they propose is interesting and challenges numerous assumptions regarding the seasonality of photosynthesis. That being said, some recent manuscripts by Wu et al. written by many of the authors (see also Kiew et al. doi.org/10.1016/j.agrformet.2017.10.022) indicate a strong vapor pressure deficit limitation to GPP that could (potentially) add quite a bit to the present manuscript given that highly statistical and empirical models are difficult to extrapolate. Investigating relationships between model residuals and VPD (or perhaps soil moisture although the authors are right in noting that its role is often over-simulated, especially given difficulties in measuring soil moisture at depth) would point toward mechanisms that other models could benefit from. Addressing the following

minor comments would in my opinion further improve an interesting manuscript.

The passages on lines 33-35 in the Abstract are self-contradictory. Please reconcile.

The intro to line 37 in the paper is a disappointment given the Amazon's central role in global heat and moisture transport and global climate teleconnections. The climate system is about energy, not just carbon. Please re-write.

The Introduction is otherwise well-written and nicely justified.

It would help the reader to justify the following passage using data on line 112-3: However, the interannual variability and trend remained the same regardless of the choice of u*Th

Note inconsistencies in italicizations between equations and text for example in lines 165-6.

192 and elsewhere: add a space between the number and the unit (in this case mm). See https://physics.nist.gov/cuu/Units/checklist.html

In paragraph 188, the definition of the dry season was a bit curious with less than 50 mm per 'half-month' of 3 or more 'semi-monthly' periods with low precipitation. Is this a running 'half-month'? Is the dry versus wet season in the Amazon not more consistently defined for people to extend the line of reasoning forwarded by this manuscript to other regions?

Line 204/205 needs a reference. Many modeling assumptions like this could benefit from more references to help the reader understand the decisions that went into model selection.

I am very surprised that the nice manuscript by Wu et al. (https://doi.org/10.1111/gcb.13509) is not cited in the present manuscript, particularly given their findings regarding diffuse radiation and vapor pressure deficit as important controls over GEP and of course the rather large overlap in authorship

between this paper and the present manuscript.

I agree on line 218 that GEE 'represents the lowest-parameter approximation of a direct measurement, but a brief explanation for readers less familiar with eddy covariance (or readers who use the eddy covariance technique but are less familiar with its limitations) would be helpful.

Qualifiers like 'strong' on line 225 and elsewhere can be avoided (and on that note of course NEE has a strong diurnal cycle). 'precisely quantified' on 369 is another example. And 'surprising' on 428. It may not have been a surprise to the forest.

On 290 do not use the * for multiplication as shorthand, this means complex conjugate (see also Fig. 5).

The material on line 312 doesn't belong in a supplement in my opinion as the seasonal patterns of RE and GEE are important to the modeling effort.

'best of a statistical model's ability' on line 324 is colloquial and probably doesn't hold for any scientific manuscript of reasonable length.

339 and elsewhere: did 2002 have anomalously high VPD? (see also the paragraph beginning line 436).

Why is 'Fig. 2b' bolded on line 356?

382: could it be shown that the hypothetical model would not add explanatory power or is this just assumed?

What is Wu et al. 2016a? This is not in the references.

Regarding the 2002, is it possible that disturbance due to tower construction may have impacted NEE? I've seen results from a few towers where there seems to be some initial transient effect on C fluxes, not that the tower wasn't constructed carefully.

Table 1: uncertainty estimates should be presented with parameter estimates. (see

also Table 2).

Figure 1: I can't help but be surprised that a forest can continuously lose C to the atmosphere, but I've seen it in other tropical forests as well when measured using the eddy covariance technique. Per earlier work by the author and team, I wonder if ustar filters are appropriate for tropical forests although trying alternate filters like sigma_w (see papers by Jocher et al.) don't seem to change things in my experience.

Avoid red (or red-ish) and green together in Fig. 7b.
* * *

---

## Author Comment (AC1) · 23 Jul 2018

(RC1) This paper investigates the influences of climate, phenology, and disturbance to NEE across various timescales in an evergreen Eastern Amazon forest using a statistical model to represent eddy covariance-derived NEE. What I concerned is the research questions are not clear and lack of storyline through the manuscript. I am curious to know what environmental drivers are responsible for the hourly, seasonal and interannual variability of NEE, respectively, and how many are the relative contributions to the variability of NEE between exogenous changes and endogenous biophysical changes.

(AC1) We thank the reviewer for the suggestions and we restructured the manuscript to make the narrative clearer and more coherent.

[Figure]

We have reframed our motivating research questions in the last paragraph of the introduction, to respectively reflect the exogenous and endogenous influences upon NEE which we discuss throughout the paper (1) meteorology at hourly to interannual timescales (2) phenology at seasonal timescales, and (3) disturbance at interannual timescales, specifically those caused by droughts.

We reordered the subsections in our results and discussion to in turn reflect the reframing of our introduction. Specifically, the we re-numbered the subsections of Results Section 3.4, which concerns our empirical modeling of NEE on various timescales, dedicating subsections to quantify the influences upon modeled NEE on hourly timescales (3.4.1) and seasonal timescales (3.4.2), and also discusses the extent to which modeled meteorology and phenology explain NEE on interannual timescales (3.4.3). For each of these sections, we explicitly quantify the relative contribution of each influence upon NEE.

Our Discussion section now more closely mirrors the timescale-based structure of the Results section as well, with sections devoted to discussing the influences upon the hourly and seasonal variability (4.1) and the interannual variability (4.2). Importantly, the sections of the Discussion that the reviewer found disconnected, those concerning droughts and disturbances, were moved into subsections of the interannual variability (4.2.1 and 4.2.2, respectively). This gives the discussion of these phenomena a more fitting home: as potential explanations for the patterns seen in annual NEE.

(RC1) Throughout the abstract, it should be more quantitative in nature. For example, how many are the relative influences of climate, phenology, and disturbance to NEE across various timescales?

(AC1) We agree that the abstract would benefit from additional quantitative information concerning the relative influences upon NEE. We included information regarding the extent to which phenology influences NEE (1% and 26% on hourly and monthly timescales respectively) in our revised abstract, explaining that the rest of the modeled

variability on these timescales is due to meteorology alone. Our abstract discusses and quantifies the extent to which meteorology and disturbance affect annual NEE at this site.

(RC1) Line 46-49 please cite the references for each driver respectively.

(AC1) We considered separating references by drivers, but most of the studies we cited examine and quantify the effects of multiple drivers simultaneously.

(RC1) Line 81-82 did Urbanski et al. (2007) answer this question? Please double check and use the same term: R/RE.

(AC1) Urbanski et al. answered this question as it applied to the Harvard Forest, a seasonal temperate mid-latitude mixed deciduous forest. Additionally, we corrected this term to RE, consistent with the abbreviation throughout the rest of the manuscript.

(RC1) Line 80-89 what are relationships among these three questions? Each question looks individual.

(AC1) We revised them to clarify the connections amongst them. Our three research questions now reflect meteorology, phenology, and drought respectively, and our concluding sentence in this paragraph summarizes them in tandem (see first comment above for details).

(RC1) Line 94 There was 1918 mm of annual rainfall. Why did it have a 5 month long dry season?

(AC1) We clarified the dates of the dry season and the definition which we and Hutyra et al. (2007) use: mean monthly precipitation of less than 100 mm.

(RC1) Line 91-98 Please describe more about the climate condition (temperature), species composition, vegetation/soil type, forest age, water table depth etc.

(AC1) We have added more site-specific information as requested.

[Figure]

(RC1) Line143-145 did you create the correlation and verified each other for these three data sets?

(AC1) We now clarify that the correlation between the monthly rainfall datasets for the years 2001-2012 was R2=0.88. We clarify that additional information about the robustness of the meteorological data sets can be found in Longo (2014).

(RC1) Line 165-170 did you make any validation for predicted CWD from the box model?

(AC1) We now clarify that a validation of the model was made against our second and final measurement of CWD in 2012.

(RC1) Line 172 what is the low-parameter empirical model?

(AC1) We specify that this refers to the model of Eq. 2.

(RC1) Line 222-225, these sentences should be placed in Methods, rather than Results.

(AC1) We removed these sentences from the results and placed them the Methods Section 2.3.

(RC1) Line 227-230 since Hutyra et al. (2007) has reported these results, what is the sense of your results here? Same problem with line 238-239.

(AC1) We specify that our slightly different gap-filling methods (Section 2.3) did not create a discrepancy within 95% confidence intervals of random measurement error, between our annually integrated NEE and that of Hutyra et al. (2007), which gives us more confidence that the post-2008 carbon balance results are consistent with previous analyses.

(RC1) Line 271 Were there any disturbance for 1998-2000?

(AC1) We did not measure CWD in 1998-2000. However, we added the caveat that

we assumed that the disturbance occurred in 1998, because 1999 and 2000 were not characterized by below-average rainfall, to Results Section 3.3, to clarify that we could only infer but did not measure that a disturbance occurred in this year. We provide additional recent evidence that drought events are accompanied by increased mortality and canopy turnover rates, signifying disturbance (Leitold et al., 2018).

(RC1) Line 762 how about the dry seasonal? Why only 9 days, rather than 90 days?

(AC1) We chose a small sampling of day from the wet season as an example to highlight the variability between days of different cloudiness. Dry season days tend to be consistently sunny and high-uptake, and therefore an example time series from the dry season would not exemplify variability between days. We specify in the Results Section 3.4.1 "Modeled hourly variability frequently captured the difference in magnitude in NEE between high and low uptake events". We chose 9 days for our example time series because many more would make the time series figure too crowded and make it hard for the reader to notice the variability between hours and days.

(RC1) The results section is pretty confusing. Please organize the idea logically. As your title: 'Carbon Exchange in an Amazon Forest: from Hours to Years', what environmental drivers are responsible for the hourly, seasonal and interannual variability of NEE, respectively? How many are the relative contributions to the variability of NEE between exogenous changes and endogenous biophysical changes?

(AC1) See our response above to the first comment for substantial revisions made to the organization of the Results section.

(RC1) How did you define the hourly, seasonal and interannual variability of NEE?

(AC1) Our usage of the term "variability" in NEE reflects the standard within a long history of applied statistical research: deviations of our dependent variable around its mean value. Throughout the paper and for the various timescales examined in our analysis, our use of the term "variability" is consistent with this definition; we therefore

did not see the need to define it explicitly within the manuscript.

(RC1) Line 325 what is the R2 between modeled and observed annual NEE?

(AC1) We added to this paragraph in Section 3.4.3 that the correlation between modeled and measured yearly NEE was low (R2 = 0.17; p = 0.37) owing to the 2002 outlier; if 2002 is excluded, the correlation is high and significant (R2 = 0.81; p = 0.014). Much of our discussion section is then devoted to discussing why 2002 was an outlier with respect to other years.

(RC1) It is also difficult to follow the Discussion section. What are the relationships between these four subtitles? They look pretty individual. Did you discuss the hourly and seasonal variability of NEE? Why don't you focus on explaining your results instead of talking about implications so much? The reader is really curious about your findings.

(AC1) See our response above to the first comment for substantial revisions made to the organization of the Discussion section.

———————————————————————

---

## Author Comment (AC2) · 23 Jul 2018

(RC2) Hayek and others explore the net ecosystem exchange of carbon and its components over multiple years in a forest in the eastern Amazon. The simple model that they propose is interesting and challenges numerous assumptions regarding the seasonality of photosynthesis. That being said, some recent manuscripts by Wu et al. written by many of the authors (see also Kiew et al. doi.org/10.1016/j.agrformet.2017.10.022) indicate a strong vapor pressure deficit limitation to GPP that could (potentially) add quite a bit to the present manuscript given that highly statistical and empirical models are difficult to extrapolate. Investigating relationships between model residuals and VPD (or perhaps soil moisture although the authors are right in noting that its role is often over-simulated, especially given difficulties in measuring soil moisture at depth) would point

toward mechanisms that other models could benefit from. Addressing the following minor comments would in my opinion further improve an interesting manuscript.

(AC2) We agree that VPD can potentially exert a strong influence on NEE at timescales discussed in this manuscript. We actually included this variable in an early version of the analysis and manuscript, originally excluding it from this analysis. We incorporated some of these results back into our revised manuscript for reasons explained below.

We ran a monthly BIC for model selection on the annual average fit of our model and others on a monthly timescale, including models that had representations of the influence of VPD and diffuse radiation upon LUE, both together and separately. The model selection method accepted these variables for the hourly model fit, but rejected them for the monthly fit, implying that the additional independent variables did not add explanatory power at monthly and longer timescales. We now include this information in the Methods Section 2.6 and present the BIC scores Results Section 3.4.2.

These results are consistent with the results of the Wu et al. (2016b) manuscript to which the reviewer refers. Our conclusions were that many of these exogenous variables add explanatory power (in that case, to GEP) at hourly and daily timescales, but on timescales of months or longer, they become increasingly outweighed in effect size and statistical significance endogenous ecosystem changes. We therefore included a reflection on these results in the hourly and seasonal NEE Discussion Section 4.1 of our revised manuscript.

Furthermore, regarding interannual variability, VPD and diffuse radiation did not explain the 2002 anomaly, the subject of our discussion regarding legacy effects of the 1998 drought. In the BIC-rejected model containing both VPD and CI, the positive NEE/GEE anomaly remained. The difference between the 2002 model-data annual residual NEE from this VPD-containing model and our model was statistically insignificant. We also examined whether VPD or diffuse radiation were anomalously high or low, respectively, in 2002, causing decreased LUE and leading to lower GEP. Both annual mean VPD

and diffuse radiation in 2002 lied within their decadal range.

Per the reviewer's suggestion to include a discussion of these variables for additional mechanistic insight, we discuss the anomaly analysis from our higher-parameter VPD and diffuse radiation model in Discussion Section 4.2 of our revised manuscript, and demonstrate that these variables did not significantly change the 2002 positive anomaly in NEE/GEE in our newly added supplemental Fig. S3.

(RC2) The passages on lines 33-35 in the Abstract are self-contradictory. Please reconcile.

(AC2) We removed reference to the 2005 and 2010 droughts specifically, which did not affect this site, and clarify in lines 31-32 that the 1998 drought did affect this site. We discuss the lack of impacts of the 2005 and 2010 droughts upon this site in the Discussion Section 4.2.1.

(RC2) The intro to line 37 in the paper is a disappointment given the Amazon's central role in global heat and moisture transport and global climate teleconnections. The climate system is about energy, not just carbon. Please re-write.

(AC2) We agree with the reviewer that energy and matter are both exchanged in large quantities by Amazon forests. We revised the introductory sentence to reflect this.

(RC2) The Introduction is otherwise well-written and nicely justified.

(AC2) We thank the reviewer for this assessment. We believe our introduction has been made even stronger by incorporating feedback from RC1.

(RC2) It would help the reader to justify the following passage using data on line 112-3: However, the interannual variability and trend remained the same regardless of the choice of u*Th

(AC2) We agree that this tendency in the data is important to highlight. We now refer the reader to Saleska et al., (2003), where this tendency for the variability and trend

to not be affected by the choice of u* filter can be seen for the first three full years of carbon exchange data for our site.

(RC2) Note inconsistencies in italicizations between equations and text for example in lines 165-6.

(AC2)We thank the reviewer for highlighting this discrepancy and have corrected it.

(RC2) 192 and elsewhere: add a space between the number and the unit (in this case mm). See https://physics.nist.gov/cuu/Units/checklist.html

(AC2) We have corrected this mistake.

(RC2) In paragraph 188, the definition of the dry season was a bit curious with less than 50 mm per 'half-month' of 3 or more 'semi-monthly' periods with low precipitation. Is this a running 'half-month'? Is the dry versus wet season in the Amazon not more consistently defined for people to extend the line of reasoning forwarded by this manuscript to other regions?

(AC2) Our definition of 50 mm per half-month is consistent with previous work on Amazon forest seasonal variability, which have collectively defined the dry season as 100 mm/month. We selected half-months to get higher resolution for the seasonal onset. Our definition of the wet season onset was unique but still consistent with the common 100 mm month-1 definition. We selected this definition because it allows for sporadic dry season downpour and ensures that there is not more than one dry season per year. Seasonality in tropical forests is typically defined by the mean seasonal cycle, whereas we were interested in interannual variability in the onset of the wet and dry seasons among various years. We added an additional line in the Methods section to explain that these half-month sums of rainfall were consistent with the common literature definition of full-month sums and provide reference.

(RC2) Line 204/205 needs a reference. Many modeling assumptions like this could benefit from more references to help the reader understand the decisions that went

into model selection.

(AC2) We added a reference to the Wu et al. GCB manuscript this line and clarified that its inclusion was insignificant after controlling for other variables in the model.

(RC2) I am very surprised that the nice manuscript by Wu et al. (https://doi.org/10.1111/gcb.13509) is not cited in the present manuscript, particularly given their findings regarding diffuse radiation and vapor pressure deficit as important controls over GEP and of course the rather large overlap in authorship between this paper and the present manuscript.

(AC2) See our response to the first RC2 comment.

(RC2) I agree on line 218 that GEE 'represents the lowest-parameter approximation of a direct measurement, but a brief explanation for readers less familiar with eddy covariance (or readers who use the eddy covariance technique but are less familiar with its limitations) would be helpful. Qualifiers like 'strong' on line 225 and elsewhere can be avoided (and on that note of course NEE has a strong diurnal cycle). 'precisely quantified' on 369 is another example. And 'surprising' on 428. It may not have been a surprise to the forest.

(AC2) Per the reviewer's suggestions, we changed these lines by omitting these imprecise qualifiers and provide additional quantitative information for the GEE estimate and the reversal of the trend in annual NEE.

(RC2) On 290 do not use the * for multiplication as shorthand, this means complex conjugate (see also Fig. 5).

(AC2) We thank the reviewer for pointing out this mistake. We removed the parameter value from this line and corrected it in the caption of Fig. 5.

(RC2) The material on line 312 doesn't belong in a supplement in my opinion as the seasonal patterns of RE and GEE are important to the modeling effort.

(AC2) We agree that the seasonal patterns of carbon exchange are relevant to modeling efforts. This figure, however, concerns the interannual variability, and we believe that much of the information that it contains is already summarized aptly by figure 7b. We included this figure in the supplement for readers who were additionally curious about how annually averaged model-data residuals compare to the range in annual means for both the data and the model.

(RC2) 'best of a statistical model's ability' on line 324 is colloquial and probably doesn't hold for any scientific manuscript of reasonable length.

(AC2) We removed this phrase from the sentence.

(RC2) 339 and elsewhere: did 2002 have anomalously high VPD? (see also the paragraph beginning line 436).

(AC2) For the material in these sections, we now include discussion of VPD in Discussion Section 4.2 per the reviewer's suggestions in the first RC2 reviewer comment.

(RC2) Why is 'Fig. 2b' bolded on line 356?

(AC2) We corrected this mistake.

(RC2) 382: could it be shown that the hypothetical model would not add explanatory power or is this just assumed?

(AC2) We ran a monthly BIC for model selection on the annual average fit of our model and a spate of others, including a model that had a higher-parameter representation of phenology. The BIC score was in fact higher (less negative) for this model, implying that the additional parameters did not add explanatory power. We left this part of the analysis out of our methods, results, and discussion for the sake of brevity, being extraneous to the more relevant results. We changed this section, simplifying the discussion of phenology so as not to allude to this alternate parameterization, stating "Our single mid-year parameter simplistically up-shifts the trough in a more continuous seasonal oscillation between low and high LUE (Fig. 5) because we lacked independent

variables explaining the seasonal oscillation."

(RC2) What is Wu et al. 2016a? This is not in the references.

(AC2) We corrected this inconsistency. Wu et al. 2016a and Wu et al. 2016b are now separate references.

(RC2) Regarding the 2002, is it possible that disturbance due to tower construction may have impacted NEE? I've seen results from a few towers where there seems to be some initial transient effect on C fluxes, not that the tower wasn't constructed carefully.

(AC2) We agree with the reviewer that we cannot rule out this or other measurement artifacts, such as those caused by tower construction creating a lower initial flux footprint than that assumed in typical eddy covariance measurement systems. We provide this important caveat in Section 4.2 of the revised manuscript, but qualify it by additionally clarifying that tower construction was completed almost a year before the measurements we used, with preliminary data collection occurring during 2001 (Saleska et al., 2003), potentially allowing time for transient effects to equilibrate.

(RC2) Table 1: uncertainty estimates should be presented with parameter estimates. (see also Table 2).

(AC2) We now present 95% confidence intervals alongside mean parameter estimates in Tables 1 and 2.

(RC2) Figure 1: I can't help but be surprised that a forest can continuously lose C to the atmosphere, but I've seen it in other tropical forests as well when measured using the eddy covariance technique. Per earlier work by the author and team, I wonder if ustar filters are appropriate for tropical forests although trying alternate filters like sigma_w (see papers by Jocher et al.) don't seem to change things in my experience.

(AC2) We found that similar filters that are turbulence-based indeed did not change our results, and in fact their use suggested that even more C was lost to the atmosphere when we used that approach. We included results from our alterative nighttime bias
correction in Fig S2. We now include a reference to Fig. S2 in the caption of Fig. 1.

(RC2) Avoid red (or red-ish) and green together in Fig. 7b.

(AC2) We changed these colors to be monochromatic in black and gray, respectively.

———————————————————